# What are Deaf sign language users' experiences as patients in healthcare services? A scoping review

**Katherine D. Rogers**[1]*, **Karina Lovell**[1], **Peter Bower**[2], **Christopher J. Armitage**[3,4,5], **Alys Young**[1,6]

1 Division of Nursing, Midwifery and Social Work, University of Manchester, Manchester, United Kingdom,
2 Division of Population Health, Health Services Research and Primary Care, University of Manchester, Manchester, United Kingdom, 3 Manchester Centre for Health Psychology, Manchester, United Kingdom,
4 Division of Psychology and Mental Health, University of Manchester, Manchester, United Kingdom,
5 NIHR Greater Manchester Patient Safety Translational Research Collaboration, United Kingdom,
6 Centre for Deaf Studies, University of the Witwatersrand, Johannesburg, South Africa

* Katherine.rogers@manchester.ac.uk

## Abstract

### Background

Deaf people who use a signed language experience poorer physical and mental health outcomes and inequalities in access and delivery of health services.

### Objectives

A scoping review was conducted to identify and synthesise current knowledge on the perspectives of Deaf people and their experience of healthcare.

### Search strategy

Databases PsycINFO, PubMed, Web of Science, and CINAHL were used for this review.

### Inclusion criteria

Any studies internationally of any design that involve Deaf signing populations that reported on patient experience within healthcare settings from a Deaf perspective were included in the review.

### Data extraction and synthesis

The abstract, title, and initial screening was followed by full text screening completed by two screeners independently. The extracted data included descriptive data and study findings. The Crowe Critical Appraisal Tool (CCAT) was applied, and findings were summarised using narrative synthesis.

### Main results

Across the 51 included papers, problems with language, communication and interaction featured prominently. Failure to meet Deaf people's needs has adverse impacts; examples

**Data availability statement:** All data are in the manuscript and/or Supporting information file.

**Funding:** This study is funded by the National Institute for Health and Care Research (NIHR) [KR is funded by the NIHR Post-Doctoral Research Fellowship (PDF-2018-11-ST2-004)]. KL is supported by the NIHR Applied Research Collaboration Greater Manchester. PB is supported by the NIHR Applied Research Collaboration Greater Manchester, NIHR School for Primary Care Research and the NIHR Policy Research Unit in Healthy Ageing. CA is supported by the NIHR Biomedical Research Centre, the NIHR Greater Manchester Patient Safety Research Collaboration, and the NIHR Applied Research Collaboration Greater Manchester. AY is supported by the Biomedical Research Centre. The views expressed are those of the author(s) and not necessarily those of the NIHR or the Department of Health and Social Care. The funders had no role in study design, data collection and analysis, decision to publish, or preparation of the manuscript.

**Competing interests:** The authors have declared that no competing interests exist.

of these include negative emotional state, disempowerment through lack of knowledge, and lack of confidence in healthcare systems.

## Discussion and conclusions

The review uniquely focused on data generated from Deaf people regarding their experience, rather than third party commentary. It confirmed a less than optimal Deaf patient experience, clinical impacts of poor experience, and negative patient and healthcare systems outcomes. This supports the case for a reliable and valid measure in a signed language to capture Deaf patients' experience in healthcare.

## Introduction

This scoping review concerns Deaf adult patients who are sign language users. There are strong cultural communities of Deaf people throughout the world. These populations are distinguished from the larger numbers of people who experience hearing loss or choose to use spoken language without any affiliation with Deaf communities and for whom being Deaf is not a cultural marker of identity [1]. Signed languages, such as British Sign Language (BSL), are complete visual languages employing space, location, handshape and movement to convey meaning [2]. They are not visual representations of the spoken/written language of whichever nation and do not have a written form. Deaf Communities are minority linguistic communities and are generally highly geographically dispersed. The Deaf Community of BSL users in the UK for example, is made up of around 87,000 people [3].

In comparison with hearing populations, Deaf people who use a signed language experience poorer mental health [4], and poorer physical health (e.g. obesity; cardiovascular disease; and diabetes mellitus) [5], as well as barriers in accessing health and care services, which are ill-suited to their cultural-linguistic requirements and preferences [6]. The resulting health inequalities have significant impacts – poorer quality of life, increased morbidity, and earlier mortality – although the pathways to these outcomes is less well researched [7]. The impacts of health inequalities can also be seen in a wider social context, for example, in the cost to the economy of inappropriate support for Deaf people who are experiencing mental health difficulties [8] and the loss to the workforce resulting from sickness [9].

In the UK, the NHS (National Health Service) Institute for Innovation and Improvement [10] defines patient experience as "what the process of receiving care feels like for the patient, their family and carers" (p.8). Valuing patients' experiences has become important in improving healthcare and health outcomes [11]. Deaf populations have historically been excluded from research, and as such they experience inequalities in health outcomes and access to healthcare [12]; therefore whether healthcare providers value Deaf patients' experience is questionable. The NICE guideline on "Patient experience in adult NHS services" [13] states that The Equality Act 2010 "provides an important legal framework which should improve the experience of all patients using NHS services" (p.5). However, there is little consistent and valid data from Deaf patients on their healthcare experience, largely because the existing *measures* for capturing patient experience are not necessarily suitable for Deaf people. This is in part an issue of language/translation, but it is also more deeply an issue of culture and perception: what is it that this cultural-linguistic community values and requires for a satisfying patient experience? Are current measures of experience able to capture that? A scoping review was conducted to identify and synthesise current knowledge on the perspectives of Deaf people and their experience of healthcare. This synthesis is a necessary

precursor to identifying and prioritising topics to be included in an appropriate patient experience measure for Deaf signers.

### Objectives and research question

Searches of PROSPERO, the Cochrane Library and the NIHR Journals Library revealed that there are no pre-existing, nor current, systematic reviews on the topic of Deaf people's experience of health services. As this review draws upon qualitative and quantitative work, the PICo approach (Population, (Phenomena of) Interest and Context) was employed to guide the question formation. The research question is as follows: "What are Deaf sign language users' experiences as patients in healthcare services?"

## Methods

This scoping review, a review that aimed to rapidly identify the breadth, depth, and type of literature available related to the key concepts, was undertaken as the first stage in a future study that will produce a Patient Reported Experience Measure (PREM) in BSL, initially validated for Deaf people living in the UK. Such a development might have wider international relevance: although Deaf communities and signed languages vary around the world, Deaf people are also transnational people with a core of common cultural values, priorities and experiences that extend beyond national identities. The method for the scoping review followed Arksey and O'Malley's [14] methodology framework for scoping studies and followed the PRISMA-ScR reporting guidelines [15]. This comprised five stages: (i) identifying the research question; (ii) identifying relevant studies; (iii) study selection; (iv) charting the data; and (v) collating, summarising, and reporting the results.

### Protocol and registration

The review protocol is available on the INPLASY website (registration number: INPLASY202210102, URL link: https://inplasy.com/inplasy-2022-1-0102/).

### Eligibility criteria

Any studies using any designs that involve Deaf signing individuals or populations worldwide were included in the review. Any studies with Deaf populations who did not use sign language, or those who were predominantly spoken language users, were excluded as they were not applicable to Deaf signing populations. This included the studies in which it was not possible to identify the data from Deaf sign language users separately from that of Deaf people who do not use sign language. Only data generated from Deaf sign language users were eligible; data 'about' Deaf patients written by others without the inclusion of data from Deaf people themselves were excluded. Included studies could be published as peer reviewed articles, book chapters, or research reports, or in grey literature including policy documents and practice guidance. Unpublished sources, online blogs/information sites and social media were excluded.

### Information sources

The bibliographic databases searched included PsycINFO, PubMed, Web of Science, and CINAHL. Forward citation searches, from the relevant reference lists, were also conducted. No data or supporting information were obtained from another source other than available in the published sources. There were no missing data. All were searched up to 25th February 2024.

## Search

Three groups of key words were searched for, related to the Deaf populations, patient experience, and healthcare (please see Table 1 for more details). The final results were then moved to EndNote, ready for the next step of the eligibility screening search.

## Selection of sources of evidence

The selection of the studies included in the review involved two stages: (i) title and abstract screening; and (ii) full text screening assisted by Rayyan software. There were two reviewers at each stage (Authors KDR and KL) and a third (Author AY) resolved any disagreements. Reasons for exclusion at the second stage of screening were recorded.

## Data charting process

Following stage two screening, relevant data from each item were extracted and recorded in a bespoke Microsoft Excel spreadsheet, including information such as study design, study sample, and results. A quality assessment for critical appraisal was recorded in a separate Excel spreadsheet.

## Data items

Extracted descriptive data included author(s), year of publication, publication type, country, participant characteristics, and setting. Extracted outcome data included study design, methods, sample, analysis approach, intervention (if any), comparisons (if using the control group), outcome data/results (statistical data or results from qualitative data). Detailed information regarding the descriptive data in each study, and the findings of each of the studies, were presented in the data extraction table in MS Excel.

## Critical appraisal of individual sources of evidence

Although not strictly required for a scoping review, a quality assessment of identified items was undertaken using the Crowe Critical Appraisal Tool (CCAT) alongside the CCAT user guide (https://conchra.com.au/2015/12/08/crowe-critical-appraisal-tool-v1-4/). The CCAT has eight categories (22 items in total) to appraise and score: (i) Preliminaries; (ii) Introduction; (iii) Design; (iv) Sampling; (v) Data collection; (vi) Ethical matters; (vii) Results; and (viii) Discussion. The CCAT was considered suitable for this review as it includes both quantitative and qualitative studies, and the reliability and validity of CCAT have been confirmed [16].

## Synthesis of results

Following Popay et al. [17], narrative synthesis was used to organise and present evidence. It was preferred because of the focus on patient experience, which involves a form of storytelling

**Table 1. Search terms used.**

| # | |
|---|---|
| S1 | (deaf* OR "hearing impair*" OR "hearing loss" OR "hard of hearing" OR "DHH" OR culturally Deaf) AND ("sign* language" OR signing) |
| S2 | (patient*) AND (experience OR satisfaction OR cent* OR activation OR feedback OR perspective OR opinion) |
| S3 | (health* setting OR healthcare OR health care OR health* service OR health* professionals OR health* personnel) |
| S4 | (S1 AND S2 AND S3) |

in which the characteristics of those providing data and how they see themselves in relation to health systems may be as significant as any outcome data.

## Results

### Selection of sources of evidence

After duplicates were removed, a total of 537 papers were identified from the database searches and 51 were included in the review. S1 Table on full text articles assessed for eligibility with the list of reasons for articles that have been excluded. See Fig 1 for the PRISMA flow diagram.

### Characteristics of sources of evidence

S2 Table on study characteristics provides details of the study characteristics. The study characteristics table (S2 Table) only refers to studies that were eligible to be included and were in the review. Countries included in the review were: USA (n = 29); UK (n = 5); Brazil (n = 4); Australia (n = 3); Spain (n = 2); South Africa (n = 2); Nigeria (n = 1); Malaysia (n = 1); Canada (n = 1); Germany (n = 1); Italy (n = 1); New Zealand (n = 1). Journal article (n = 44)

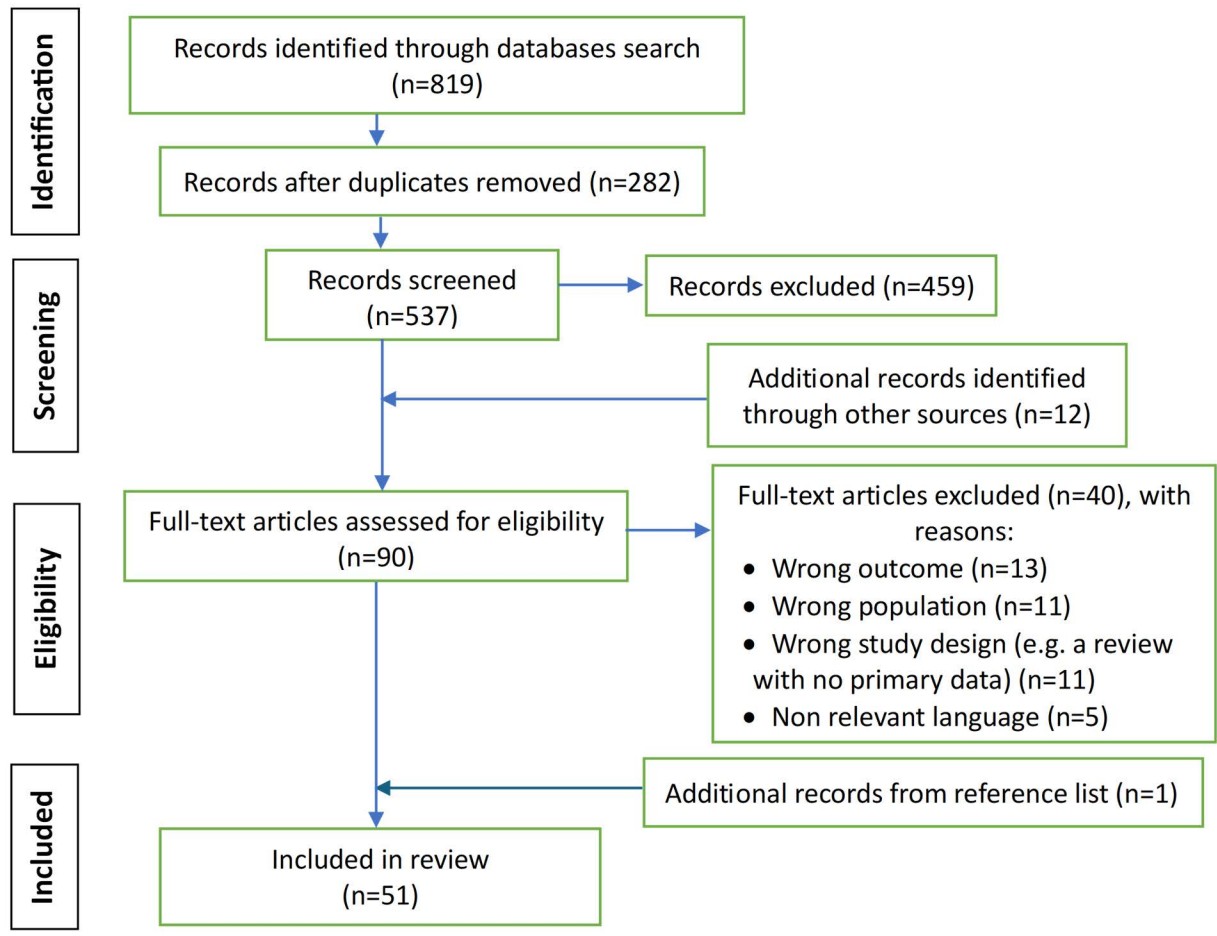

**Fig 1. PRISMA flow diagram: the findings from the searches.**

is the major publication type in this scoping review when compared to other publication types (book chapter = 1; report = 3; PhD thesis = 2, and master's thesis = 1). Most study participants were female (12 studies only recruited females); however, gender was not reported in 5 studies. Date range of included items was 18 to 90 plus years old. The majority of the studies were qualitative, with 32 (62.7%) being qualitative (n = 22 interview, n = 12 focus group, and n = 2), 12 (23.5%) used a mixed methods approach, and 7 (13.7%) were quantitative.

## Critical appraisal within source of evidence

The results of the critical appraisal are reported in Table 2. Of 51 included items, 13 scored 70-99%; 15 score 60-69%; 14 50-59%; 7 scored 40-49%, and 3 scored 30-39% indicating they were of a poor quality.

## Narrative synthesis

The predominant issues across the 51 papers were language, communication, and interaction, but these were manifest in different aspects of the patient experience with different kinds of consequences, both individual (centred on the patient) and structural (centred on the health care delivery system). In what follows we highlight the nature of the language, communication, and interaction problems experienced, and the consequences for the patient's management of their illness and health outcomes, the quality of the clinical encounters and the ability of systems to learn and respond to Deaf people's needs in a culturally competent way.

**Experiences and consequences of language, communication and interaction needs not being recognised or met.  Language access:** Many healthcare providers who treat Deaf people do not share the same language as the Deaf individual (i.e. they are unable to use sign language). Furthermore, it has been reported not only that providers have failed or refused to provide interpreters (e.g. ID002, ID004, ID006, ID036), but additionally that they do not understand their responsibilities for providing an interpreter (ID016). Consequently, effective communication without the use of interpreters was poor. Many patients struggled with lip-reading, in part because they were unable to see the clinician's face, which is crucial for picking up non-auditory cues as well as for purposes of lipreading (ID016). Some communicated through passing written notes between professional and patient, but this is problematic if the written/spoken language of the country was not their first or preferred language or literacy was poor (e.g. ID001, ID006, ID020, ID025, ID027, ID038, ID040, ID043, ID048). In addition, Deaf people reported having to advocate for their own accessibility, such as the need for a sign language interpreter (ID010, ID020, ID021, ID032, ID037, ID040) as part of their appointment with the healthcare provider.

There are two main impacts of the communication barriers encountered, namely, knowledge and emotional consequences.

**Knowledge consequences:** Poor quality interactions and communication with healthcare professionals meant that patients did not fully understand their condition or the instructions they were being given regarding treatment (ID018, ID005, ID046, ID006, ID008, ID024, ID029, ID032, ID033, ID034, ID047). Patients reported leaving an appointment with minimal understanding of a diagnosis, and confusion about treatment options or follow-up appointments (ID004, ID020, ID037, ID041). This had effects on the patient's confidence in their healthcare practitioner (ID006, ID023, ID043) and their ability to manage their own condition and participate in their care (ID047). It also potentially presented a risk to health and patient safety if, for example, written discharge instructions were inaccessible (ID020, ID021) or patients were left unsure how to manage their medication (ID005, ID009, ID010, ID017, ID037, ID040, ID049). For example, instructions for frequency of inhaler use were

**Table 2. CCAT score.**

| Study ID | Authors/year | Preliminary | Introduction | Design | Sampling | Data Collection | Ethics | Results | Discussion | Total score | % |
|---|---|---|---|---|---|---|---|---|---|---|---|
| ID001 | Adigun et al (2020) [18] | 4 | 5 | 3 | 4 | 4 | 2 | 4 | 4 | 30 | 75 |
| ID002 | Anderson et al (2017) [19] | 4 | 4 | 2 | 3 | 4 | 3 | 4 | 4 | 28 | 70 |
| ID003 | Berman et al (2013) [20] | 4 | 4 | 4 | 3 | 4 | 4 | 3 | 4 | 30 | 75 |
| ID004 | Berman et al (2017) [21] | 4 | 4 | 3 | 4 | 4 | 4 | 4 | 4 | 31 | 78 |
| ID005 | Cardoso et al (2006) [22] | 2 | 4 | 2 | 2 | 3 | 3 | 4 | 2 | 22 | 55 |
| ID006 | Cerilli et al (2023) [23] | 4 | 4 | 3 | 3 | 3 | 2 | 3 | 4 | 23 | 58 |
| ID007 | Chin et al (2013) [24] | 4 | 4 | 2 | 3 | 3 | 2 | 4 | 4 | 26 | 65 |
| ID008 | Costa et al (2018) [25] | 3 | 4 | 2 | 4 | 4 | 3 | 3 | 3 | 24 | 60 |
| ID009 | Ferguson et al (2003) [26] | 4 | 4 | 3 | 2 | 4 | 2 | 4 | 4 | 27 | 68 |
| ID010 | Fernandez-Valderas et al (2017) [27] | 3 | 1 | 2 | 2 | 3 | 3 | 2 | 3 | 19 | 48 |
| ID011 | Foltz & Shank (2020) [28] | 1 | 3 | 1 | 1 | 1 | 3 | 2 | 3 | 15 | 38 |
| ID012 | Gichane et al (2017) [29] | 4 | 4 | 3 | 3 | 3 | 3 | 2 | 3 | 25 | 63 |
| ID013 | Gilchrist (2000) [30] | 3 | 4 | 3 | 3 | 3 | 3 | 3 | 3 | 25 | 63 |
| ID014 | Hubbard et al (2018) [31] | 2 | 3 | 2 | 2 | 2 | 1 | 3 | 2 | 17 | 43 |
| ID015 | Iezzoni et al (2004) [32] | 4 | 3 | 3 | 2 | 3 | 3 | 4 | 3 | 25 | 63 |
| ID016 | Jacob et al (2021) [33] | 3 | 4 | 3 | 3 | 3 | 3 | 2 | 3 | 24 | 60 |
| ID017 | Jacobs et al (2021) [34] | 4 | 4 | 4 | 3 | 4 | 3 | 4 | 4 | 30 | 75 |
| ID018 | James et al (2022) [35] | 3 | 4 | 3 | 4 | 4 | 4 | 4 | 4 | 30 | 75 |
| ID019 | James, Panko, et al (2023) [36] | 4 | 4 | 3 | 2 | 4 | 2 | 2 | 4 | 25 | 63 |
| ID020 | James, Sullivan et al (2023) [37] | 4 | 4 | 3 | 4 | 4 | 4 | 4 | 4 | 31 | 78 |
| ID021 | Kushalnagar et al (2019) [38] | 4 | 4 | 4 | 3 | 4 | 4 | 4 | 4 | 31 | 78 |
| ID022 | Kyle et al (2013) [39] | 4 | 4 | 3 | 2 | 3 | 3 | 4 | 4 | 27 | 68 |
| ID023 | Lee et al (2021) [40] | 3 | 3 | 3 | 2 | 3 | 3 | 3 | 3 | 23 | 58 |
| ID024 | MacKinney et al (1995) [41] | 3 | 2 | 3 | 3 | 2 | 1 | 3 | 3 | 20 | 50 |
| ID025 | Miller et al (2019) [42] | 4 | 4 | 4 | 3 | 4 | 4 | 4 | 4 | 31 | 78 |
| ID026 | Mussallem et al (2022) [43] | 3 | 3 | 2 | 1 | 2 | 1 | 3 | 3 | 18 | 45 |
| ID027 | Myers et al (2022) [44] | 3 | 4 | 3 | 1 | 3 | 3 | 3 | 3 | 23 | 58 |
| ID028 | Napier et al (2013) [45] | 3 | 3 | 3 | 3 | 2 | 2 | 3 | 3 | 22 | 55 |
| ID029 | Napier et al (2014) [46] | 3 | 4 | 4 | 3 | 3 | 4 | 4 | 2 | 27 | 68 |
| ID030 | O'Hearn et al (2006) [47] | 3 | 4 | 2 | 1 | 1 | 1 | 3 | 3 | 18 | 45 |
| ID031 | Oliveira et al (2015) [48] | 2 | 3 | 3 | 2 | 2 | 2 | 1 | 3 | 18 | 45 |
| ID032 | Panko et al (2022) [49] | 3 | 3 | 2 | 3 | 3 | 3 | 3 | 3 | 23 | 58 |
| ID033 | Parise (1999) [50] | 2 | 3 | 3 | 3 | 3 | 2 | 3 | 3 | 22 | 55 |
| ID034 | Pereira & Fortes (2010) [51] | 3 | 4 | 2 | 3 | 2 | 2 | 2 | 3 | 21 | 53 |
| ID035 | Pertz et al (2018) [52] | 3 | 4 | 3 | 2 | 4 | 2 | 2 | 3 | 23 | 58 |
| ID036 | Pinilla et al (2019) [53] | 2 | 3 | 2 | 3 | 3 | 3 | 3 | 3 | 22 | 55 |
| ID037 | Reeves et al (2005) [54] | 3 | 4 | 3 | 3 | 3 | 1 | 3 | 3 | 23 | 58 |
| ID038 | Rodriguez-Martin et al (2018) [55] | 4 | 4 | 2 | 3 | 2 | 4 | 3 | 3 | 25 | 63 |
| ID039 | Schniedewind et al (2020) [56] | 4 | 4 | 4 | 3 | 4 | 3 | 3 | 4 | 29 | 73 |
| ID040 | Shank & Foltz (2019) [57] | 2 | 3 | 1 | 1 | 2 | 1 | 1 | 2 | 13 | 33 |
| ID041 | Sheppard (2014) [58] | 3 | 3 | 4 | 3 | 3 | 3 | 4 | 4 | 27 | 68 |
| ID042 | Sheppard & Badger (2010) [59] | 3 | 4 | 3 | 3 | 3 | 2 | 4 | 3 | 25 | 63 |
| ID043 | SignHealth (2014) [60] | 2 | 3 | 2 | 2 | 1 | 1 | 2 | 3 | 16 | 40 |
| ID044 | Sirch et al (2017) [61] | 4 | 3 | 4 | 4 | 4 | 4 | 4 | 4 | 31 | 78 |
| ID045 | Steinberg et al (2002) [62] | 4 | 4 | 3 | 3 | 4 | 3 | 4 | 4 | 29 | 73 |
| ID046 | Steinberg et al (2006) [63] | 4 | 3 | 3 | 2 | 3 | 3 | 3 | 4 | 25 | 63 |
| ID047 | Swannack (2018) [64] | 3 | 3 | 3 | 2 | 3 | 3 | 3 | 3 | 23 | 58 |

*(Continued)*

**Table 2.** (Continued)

| Study ID | Authors/year | Preliminary | Introduction | Design | Sampling | Data Collection | Ethics | Results | Discussion | Total score | % |
|---|---|---|---|---|---|---|---|---|---|---|---|
| ID048 | Tamaskar et al (2000) [65] | 3 | 3 | 3 | 2 | 3 | 0 | 3 | 3 | 20 | 50 |
| ID049 | Witko et al (2017) [66] | 2 | 4 | 2 | 2 | 2 | 2 | 2 | 2 | 18 | 45 |
| ID050 | Witte & Kuzel (2000) [67] | 1 | 4 | 2 | 2 | 2 | 0 | 1 | 2 | 14 | 35 |
| ID051 | Yabe et al (2020) [68] | 4 | 4 | 4 | 3 | 3 | 0 | 4 | 3 | 25 | 63 |

not clear (ID009) and they did not understand the requirement to take the medication long-term (ID049). Lack of awareness of treatment options and how to access them (ID002) also impacted decision making, particularly when the Deaf person was not certain that they understood what was going on (ID034) or had sufficient opportunity to ask questions, or even be sure of what questions to ask (ID045). For some, the failure to have their language rights recognised and met was clearly seen as discrimination (ID044) as it impeded not just knowledge and understanding but full participation in their healthcare.

By contrast, patients who had experienced practitioner-patient communication through an interpreter said this enabled them to be fully informed and better able to discuss their condition with healthcare professionals (ID018), and that they valued and trusted the healthcare professional more (ID07, ID010, ID031, ID035, ID046). Having an interpreter was reported as making it easier to engage with healthcare professionals (ID014, ID032, ID048), and allowed them the opportunity to ask questions (ID014, ID032, ID037). When making an important decision about health and wellbeing, access to information, and explanations from the doctor regarding treatment, including both the benefits and the risks, was vital (ID018, ID037).

**Emotional consequences:** Experiencing communication barriers negatively affected the emotional state of the patient during clinical encounters, with many reports of frustration at the lack of communication access (ID021, ID030, ID041, ID042), and distress at having to self-advocate for their communication needs at a time when they are feeling unwell, an experience some described as traumatic (ID021, ID034). Patients felt uncomfortable and upset when a clinician carried out a physical examination or procedure and did not give a detailed explanation as to what would occur in advance (ID015). Exclusionary feelings and frustration occurred for in-patients too. For example, shift handovers and doctors' conferences with Deaf patients created anxiety for Deaf people (ID049) because they could not follow what was happening when conversations were quick and failed to include them adequately. These experiences resulted in a lack of confidence in healthcare providers (e.g. ID006, ID046, ID047) that could extend to reluctance to engage with them because of the personal distress it can cause.

**Cultural competency.** Linked to, but also distinct from, concerns about language, communication and interaction were issues of cultural competency, whether in the direct delivery of services by healthcare practitioners or in the more structural concerns of how systems of healthcare did not accommodate Deaf patients.

**Direct interaction:** Many studies reported that Deaf people felt that their healthcare providers did not demonstrate adequate cultural knowledge, competency, and skill in interactions with them (ID002, ID003, ID004, ID007, ID010, ID016, ID021, ID024, ID030, ID033, ID034, ID044, ID045, ID047, ID049, ID050) – for example, not being aware that some Deaf people struggle to understand not just a spoken language but also a written language. Consequently, assumptions that written communication can resolve communication barriers are unhelpful (ID009, ID016, ID038). Eye contact was vital for Deaf people both in terms of understanding communication (whether signed or following lipreading) but is also a cultural norm that expresses engagement and appreciation – the person is following what you are saying. Consequently, to not make eye contact was tantamount to expressing a lack of interest in them,

fostering a feeling of a lack of engagement (ID041). Deaf patients reported attitude problems too amongst healthcare professionals, such as being impatient (ID010, ID015, ID034, ID041), not having time for them nor being able to 'listen' to them, being made to feel that they are wasting the healthcare professionals' time (ID023, ID037, ID043), and being rude (ID013), including behaviour such as 'eye rolling' (ID014). Such attitudes left patients concluding that healthcare professionals disliked working with them (ID046). Additionally, they felt that they were not being respected as patients, and the fact that they want to understand and participate in their healthcare was not being acknowledged (ID006, ID020, ID016).

**Organisational and structural barriers:** On a structural level, lack of cultural competency was manifested in the unavailability of information and services, or their inaccessibility for Deaf people stemming from failure to consider how their organisation and provision excludes Deaf people. Individuals whose preferred language was sign language desired accessible healthcare information in sign language (ID0017, ID029, ID030). Gaining information that was accessible was important for reassurance and building trust (ID045). Yet, the availability of resources in sign language that allowed Deaf people to seek information related to healthcare, health literacy and the self-management of illnesses was limited compared to the availability of resources in written languages (ID040, ID041, ID043, ID045, ID049). Lack of information provided in signed languages had consequences for outreach services and public health monitoring. For example, Deaf people might be unaware of or not respond to the call for routine screening such as a mammogram or a pap smear (ID003, ID045), because they were unaware why this was recommended through no fault of their own (ID002, ID008). Although some services might be offered for free (such as cancer support or drug/alcohol support groups), often no interpreters were provided (ID030, ID045). Without access to support groups that met their cultural and linguistic needs, Deaf people reported missing out on opportunities to learn and exchange information to help manage their condition. Services specialised to meet specific health conditions were not always willing to accept a Deaf patient (ID041). Few specialist services just for Deaf people exist, with many having long waiting lists (ID002).

Indirect discrimination occurred when usual structures of service access and provision had differential effects for Deaf people. For example, reliance on spoken telecommunications to book appointments (ID016, ID023, ID024, ID041, ID043, ID41), or obtain test results (ID023, ID046, ID049, ID050) necessitated Deaf people attending in person instead, with an interpreter. Although the use of a relay sign language interpreting service online might overcome these barriers, many health personnel did not know how or were unwilling to use such systems (ID046, ID050). Access to buildings or hospital wards via a spoken language intercom system without visual access as well was an obvious barrier for Deaf people (ID049). Cultures of ordinary behaviours could be exclusionary too, such as healthcare professionals calling out the next patient's name rather than using universal means of alerting such as visual displays with names (ID010, ID016, ID033, ID043, ID050).

**Examples of good cultural competency:** At its most basic, ensuring the provision of a sign language interpreter demonstrated cultural competency because the conditions for good patient-practitioner interaction were recognised in advance. This effort was appreciated not just for the reason of meeting communication needs, but it also demonstrated respect for the patient (ID013) and results in the healthcare provider being more valued by the Deaf patient (ID045). Other positive markers of cultural competency included professionals being aware not to use inappropriate language (such as 'hearing impaired') when referring to a culturally Deaf patient (ID049), maintaining good eye contact, using helpful gestures, and alternative means of explanations such as pictures and diagrams alongside a spoken explanation (ID036). Time was also an important factor: recognising that an appointment involving a Deaf person

and an interpreter might mean more time was required than might be usual, providing the opportunity to ask questions and gain full understanding (ID006, ID043). Consistency of seeing the same healthcare provider was also mentioned (ID009, ID010, ID011, ID023), referring to the gains from seeing the same person over time who has had the opportunity to build up their awareness and understanding of how to meet a Deaf patient's communication needs rather than having to start from scratch every time with a new person. Such consistency results in better confidence in the healthcare provider and less anxiety for the patient.

**Partial or inadequate solutions being put in place to meet the needs of Deaf people.** Many healthcare professionals did not know sign language, or were not fluent in sign language, so qualified interpreters were needed for effective communication between Deaf patients and healthcare providers. Many patients emphasised the importance of *qualified* interpreters (ID004, ID005, ID006, ID007, ID013, ID014, ID017, ID019, ID020, ID025, ID030, ID031, ID036, ID037, ID038, ID042, ID045, ID046, ID051), including those with training in Deafblind interpreting (ID011). By using interpreters, it allowed *both* parties to be understood (ID014) and enabled effective mutual communication (ID001, ID018). Interpreters are not just for the Deaf person. However, what was less well understood by healthcare providers was that the process of getting an interpreter was not easy for Deaf people (ID002, ID020, ID027, ID040), especially at short notice (ID030) and waiting times for the availability of an interpreter could cause further delays (ID015, ID021, ID027). Such arrangements were an added burden for Deaf people that negatively impacted their experience as patients. Sometimes healthcare providers booked unqualified interpreters (ID043, ID047, ID050), either through ignorance or because they were the only ones available. This had several effects, including a lack of trust in the appointment and the patient being left with partial information and understanding. Some Deaf people reported that healthcare professionals did not know how to use interpreters (ID004, ID024), and in some cases were not able to distinguish between the roles of companions, caregivers, and professional interpreters (ID034).

Healthcare professionals were unaware of the risks of not using professional interpreters (ID049). Often, healthcare professionals would expect Deaf people to bring their family members to 'interpret' or 'assist' with communication (ID016, ID021, ID029, ID033, ID034). However, hearing relatives sometimes edited the interactions, not sharing a full account of the information exchanged with a doctor, which diminished the Deaf patient's autonomy (ID034). That said, there were reports that some preferred a family member to interpret rather than a professional interpreter, depending on what the appointment involved, particularly if it was a sensitive issue (ID037).

Although Deaf people valued having an interpreter, they were concerned about confidentiality because the Deaf community is small (ID002, ID021, ID036, ID038, ID041, ID042). The same can be said about specialist service providers who know sign language (ID002). Deaf people have emphasised the importance of expressing a preference for a specific interpreter of their choice (ID014, ID045) in order to mitigate some of these concerns, as the named interpreter was one they particularly trusted. Having the same interpreter consistently throughout all appointments was also valued because of the repository of knowledge built up by the interpreter, increasing their effectiveness in complex medical interactions (ID015). There is no formal recognised specialist training for interpreters working in healthcare in most countries (ID047), although it has been reported that the best communication occurs when working with medically experienced qualified interpreters (ID027, ID046).

**Remote interpreting (VRS/VRI):** With the growing availability of providing an interpreter offsite by means of a videocall, some Deaf people expressed their views on using remote interpreters. Benefits included speed of booking compared to a face-to-face interpreter (ID021). However drawbacks included healthcare providers lacking awareness and knowledge of how to use these

services (ID006, ID027, ID040), poor connectivity (ID006, ID020, ID027, ID028, ID035, ID051), difficulties in feeling connected or in participating in the conversation between healthcare providers when the screen was not clearly visible (ID027, ID028), and interpreters being unable to pick up non-verbal communication in the same way as they would in person (ID024).

It was reported that on-site interpreters increased patient-provider engagement and shared decision making, and it was felt that a remote interpreter diminished patient-centred care and the communication experience (ID021, ID032, ID051). With regard to satisfaction in the use of a remote interpreter, those with high school education were more likely to be satisfied; having regular healthcare provider interactions meant they were less likely to be satisfied with a remote interpreter; and those who felt that remote interpreting did not interfere with disclosure were more likely to be satisfied (ID022). Those who struggled with video quality issues had lower satisfaction scores (ID035).

**Complaints about interpreters:** Complaints about interpreters varied (e.g., a lack of availability, an insufficiently qualified interpreter). In one instance a professional highlighted that an interpreter had been secured when in fact this was not the case; they were told that the providers were not taking on any new patients, which the professional considered suspicious (ID039). It was reported that only 1% of the study participants made a complaint, and 11% wanted to but did not (ID037).

**Final synthesis.** Fig 2 demonstrates the conceptual map of Deaf patient's experience. Barriers to language, communication, and interaction has negative consequences on Deaf patient's experiences, clinical impact, patient and healthcare system outcomes.

## Discussion

### Summary of evidence

This review has highlighted significant barriers faced by Deaf people in terms of the availability of, access to, and suitability of healthcare services. The main issues were language, communication, and interaction, which Deaf people stated were needs that were not being met, coupled with their cultural needs not being met. The impacts of these included negative feelings (e.g. lower wellbeing, and dissatisfaction) and difficulties in engaging with healthcare practitioners, as well as negative consequences on their knowledge (e.g. not understanding about the health conditions and the aftercare). Impacts were noted on individuals and in the lack of confidence in healthcare systems (e.g. patient safety, patient-centred care, and the timeliness of care). These issues would have further negative consequences for Deaf people on their health outcomes and treatment outcomes as well as negative health related behaviours. We wish to emphasise that these and other examples of barriers to inclusion are avoidable through changes in organisational structures and cultures of behaviour. These consequences for Deaf patients are not inevitable.

Patient-centred care processes for Deaf people would need to include several aspects, such as: the recognition of Deaf people as a cultural minority population; a repository of information to tackle health and knowledge inequalities; and the monitoring of Deaf people's experience in healthcare [12]. However, this review highlights that this is not the case for Deaf people. When one considers the communication barriers between Deaf patients and healthcare professionals, there is a lack of recognition of Deaf people as a cultural linguistic minority by healthcare professionals. The consequences include difficulties for Deaf patients being able to contribute to their own care and the decision-making process. Attitudes of healthcare professionals are not limited to having positive or negative feelings towards a person but also include acknowledging a Deaf patient as a visually oriented person, respecting them, being willing to learn how to engage with them, and so forth.

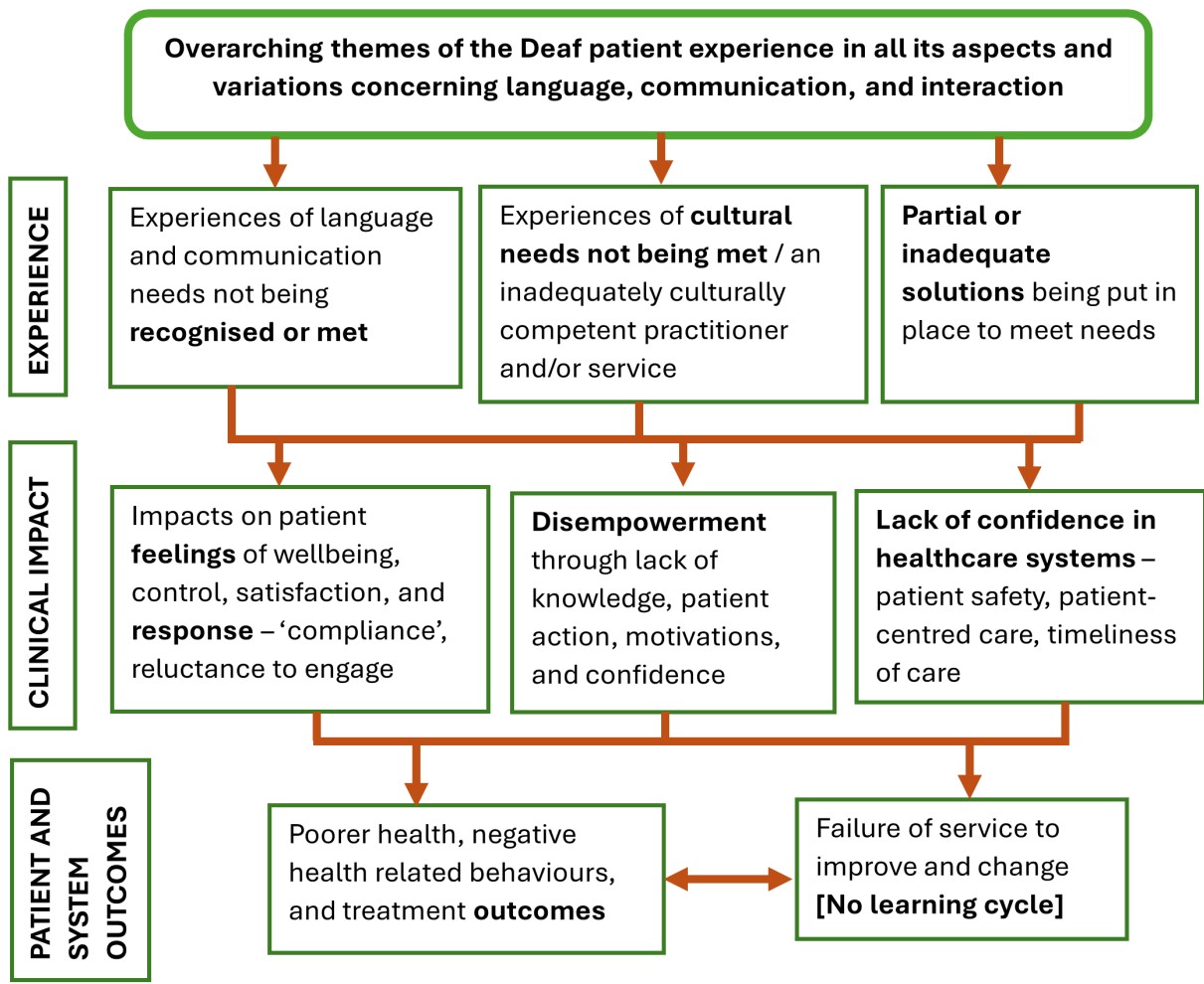

**Fig 2. A conceptual map of Deaf patient experience in healthcare.**

Although the studies included in the review reported detailing the experience faced by Deaf people, this review identified no standardised measures in sign language to obtain Deaf patients' perspectives and experience on healthcare. Existing measures that address patients' experience of healthcare include the Patient Experiences Questionnaire (PEQ) [69], and the IAPT (Improving Access to Psychological Therapies) PEQ [70]. These are strongly influenced by hearing cultural values. This is the case not just for patient experience assessments, but also more generally in relation to a range of standard instruments used to assess health, recovery and wellbeing. For example, in a previous study that included the translation and validation of the quality of life measure EQ-5D-5L, in BSL [71], responses to the question about 'problems with mobility' were influenced by Deaf people's communication considerations rather than physical restrictions (e.g., mobility might be a problem if you cannot easily tell a bus or taxi driver where you want to go). Responses to self-report measures in general are likely to be influenced by Deaf people's experience of the world, including everyday factors that hearing populations would not routinely consider [71]. Factors that are valued in interaction and communication associated with healthcare, for example, may be more highly prioritised by Deaf people than by those patients with a routine expectation of communication with professionals

in a shared language. The visual nature of Deaf people's experience of the world, whilst being fundamental to Deaf culture, is also a primary means by which the world is experienced, interpreted and understood [72]. A PREM that is culturally valid for Deaf people in their own nation's sign language and the context of healthcare is therefore required in order to better understand how healthcare organisations, systems and providers can learn from Deaf patients in order to improve the quality of healthcare for Deaf people.

### Strengths and limitations

This scoping review identified a good number of studies regarding Deaf signing populations and their experience in healthcare. Although not typically expected for a scoping review, this review carried out some quality assessment for each included study. This review only included the publications that were published in written English or Signed Language; therefore, any other languages were excluded, which may explain why many included studies were from Western countries.

### Conclusions and implications

Inequalities in obtaining patient centred care is common for Deaf people. This review has highlighted that the patient experience for Deaf people is poor, yet we do not know how poor, nor do we have any way of tracking it to identify what is the biggest issue and what needs to be improved. There is a need for a validated measure in sign language to understand the experience of Deaf people in healthcare. The implications for healthcare practitioners are to ensure delivery of high-quality healthcare services for Deaf people, whilst understanding their experiences and ensuring that Deaf people's language, communication, and interaction needs are being met. A positive patient experience for Deaf people would assist in ensuring that patient safety and effectiveness of treatment are maintained.

### Supporting information

**S1 Table.  Full-text articles assessed for eligibility (n = 91).**
(DOCX)

**S2 Table.  Study characteristics.**
(DOCX)

**S1 Checklist.  Preferred Reporting Items for Systematic Reviews and Meta-Analyses extension for Scoping Reviews (PRISMA-ScR) checklist.**
(DOCX)

**S1 Video.  What are Deaf sign language users' experiences as patients in healthcare services** ? A scoping review.
(MP4)

### Author contributions

**Conceptualization:** Katherine D. Rogers, Karina Lovell, Peter Bower, Christopher J. Armitage, Alys Young.

**Data curation:** Katherine D. Rogers, Karina Lovell, Peter Bower.

**Formal analysis:** Katherine D. Rogers.

**Funding acquisition:** Katherine D. Rogers.

**Investigation:** Katherine D. Rogers.

**Methodology:** Katherine D. Rogers, Alys Young.

**Project administration:** Katherine D. Rogers.

**Resources:** Katherine D. Rogers.

**Validation:** Katherine D. Rogers.

**Visualization:** Katherine D. Rogers, Alys Young.

**Writing – original draft:** Katherine D. Rogers.

**Writing – review & editing:** Katherine D. Rogers, Karina Lovell, Peter Bower, Christopher J. Armitage, Alys Young.

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
