## [Decision Letter · Decision Letter 0]

9 Oct 2024

PGPH-D-24-01584

What are Deaf sign language users’ experiences as patients in healthcare services? A scoping review

Dear Dr. Rogers,

Thank you for submitting your manuscript to PLOS Global Public Health. After careful consideration, we feel that it has merit but does not fully meet PLOS Global Public Health’s publication criteria as it currently stands. Therefore, we invite you to submit a revised version of the manuscript that addresses the points raised during the review process.

We look forward to receiving your revised manuscript.

Kind regards,

Vanessa Carels

Staff Editor

Additional Editor Comments (if provided):

Reviewers' comments:

Reviewer's Responses to Questions

**Comments to the Author**

1. Does this manuscript meet PLOS Global Public Health’s publication criteria ? Is the manuscript technically sound, and do the data support the conclusions? The manuscript must describe methodologically and ethically rigorous research with conclusions that are appropriately drawn based on the data presented.

Reviewer #1: Yes

Reviewer #2: Partly

2. Has the statistical analysis been performed appropriately and rigorously?

Reviewer #1: N/A

Reviewer #2: Yes

3. Have the authors made all data underlying the findings in their manuscript fully available (please refer to the Data Availability Statement at the start of the manuscript PDF file)?

Reviewer #1: Yes

Reviewer #2: Yes

4. Is the manuscript presented in an intelligible fashion and written in standard English?

Reviewer #1: Yes

Reviewer #2: Yes

5. Review Comments to the Author

Reviewer #1: The article presents a well-conducted investigation, from the research question, the methodology used, the sources of evidence, the results, to the analysis of the results and the conclusions. I have no comments and recommend that it be published.

Reviewer #2: The experiences of deaf people who need sign language assistance to access healthcare were the subject of a scoping review conducted by the authors. After reviewing 51 articles that satisfied their inclusion requirements, they compiled data showing that Deaf persons frequently felt their language and communication needs were not satisfied, in addition to their cultural demands. After doing a thorough search, the writers were able to compile their findings and publish the outcomes. The article's linguistic syntax and writing style are my main areas of concern. There were numerous misspellings and improper phrase constructions. The presentation of the article needs to be revised.

6. PLOS authors have the option to publish the peer review history of their article (what does this mean? ). If published, this will include your full peer review and any attached files.

**Do you want your identity to be public for this peer review?** For information about this choice, including consent withdrawal, please see our Privacy Policy .

Reviewer #1: No

Reviewer #2: No

---

## [Decision Letter · Decision Letter 1]

6 Jan 2025

What are Deaf sign language users’ experiences as patients in healthcare services? A scoping review

PGPH-D-24-01584R1

Dear Dr Rogers,

We are pleased to inform you that your manuscript 'What are Deaf sign language users’ experiences as patients in healthcare services? A scoping review' has been provisionally accepted for publication in PLOS Global Public Health.

Best regards,

Connie Cai Ru Gan

Academic Editor

Reviewer Comments (if any, and for reference):

Reviewer's Responses to Questions

**Comments to the Author**

1. If the authors have adequately addressed your comments raised in a previous round of review and you feel that this manuscript is now acceptable for publication, you may indicate that here to bypass the “Comments to the Author” section, enter your conflict of interest statement in the “Confidential to Editor” section, and submit your "Accept" recommendation.

Reviewer #1: All comments have been addressed

Reviewer #2: All comments have been addressed

2. Does this manuscript meet PLOS Global Public Health’s publication criteria ? Is the manuscript technically sound, and do the data support the conclusions? The manuscript must describe methodologically and ethically rigorous research with conclusions that are appropriately drawn based on the data presented.

Reviewer #1: Yes

Reviewer #2: Yes

3. Has the statistical analysis been performed appropriately and rigorously?

Reviewer #1: Yes

Reviewer #2: Yes

4. Have the authors made all data underlying the findings in their manuscript fully available (please refer to the Data Availability Statement at the start of the manuscript PDF file)?

Reviewer #1: Yes

Reviewer #2: Yes

5. Is the manuscript presented in an intelligible fashion and written in standard English?

Reviewer #1: Yes

Reviewer #2: Yes

6. Review Comments to the Author

Reviewer #1: I have reviewed the two manuscripts, the original and the one submitted in December 2024, comparatively. The authors placed the wording of the results in the past tense and changed some terms, improving the understanding of the text. I reiterate that the proposed article is in a position to be published.

Reviewer #2: (No Response)

7. PLOS authors have the option to publish the peer review history of their article (what does this mean? ). If published, this will include your full peer review and any attached files.

**Do you want your identity to be public for this peer review?** For information about this choice, including consent withdrawal, please see our Privacy Policy .

Reviewer #1: No

Reviewer #2: No
